# Effects of Heat Acclimation Following Heat Acclimatization on Whole Body Heat Exchange in Trained Endurance Athletes

**DOI:** 10.3390/ijerph19116412

**Published:** 2022-05-25

**Authors:** Yasuki Sekiguchi, Courteney L. Benjamin, Elaine C. Lee, Jeb F. Struder, Ciara N. Manning, Margaret C. Morrissey, Michael R. Szymanski, Rebecca L. Stearns, Lawrence E. Armstrong, Douglas J. Casa

**Affiliations:** 1Korey Stringer Institute, Department of Kinesiology, University of Connecticut, Storrs, CT 06269, USA; cbenjami@samford.edu (C.L.B.); jeb.struder@uconn.edu (J.F.S.); ciara.manning@uconn.edu (C.N.M.); margaret.morrissey@uconn.edu (M.C.M.); michael.szymanski@uconn.edu (M.R.S.); rebecca.stearns@uconn.edu (R.L.S.); douglas.casa@uconn.edu (D.J.C.); 2Sports Performance Laboratory, Department of Kinesiology and Sport Management, Texas Tech University, Lubbock, TX 79409, USA; 3Department of Kinesiology, Samford University, Birmingham, AL 35229, USA; 4Human Performance Laboratory, Department of Kinesiology, University of Connecticut, Storrs, CT 06269, USA; elaine.c.lee@uconn.edu (E.C.L.); uconnla@aim.com (L.E.A.)

**Keywords:** heat balance, evaporation, dry heat loss, running economy, metabolic heat production, heat adaptations

## Abstract

The purpose of this study was to examine the changes in metabolic heat production (H_prod_), evaporative heat loss (H_evap_), and dry heat loss (H_dry_), following heat acclimatization (HAz) and heat acclimation (HA). Twenty-two male endurance athletes (mean ± standard deviation; age, 37 ± 12 y; body mass, 73.4 ± 8.7 kg; height, 178.7 ± 6.8 cm; and VO_2max_, 57.1 ± 7.2 mL·kg^−1^·min^−1^) completed three trials (baseline; post-HAz; and post-HA), which consisted of 60 min steady state exercise at 59 ± 2% velocityVO_2max_ in the heat (ambient temperature [T_amb_], 35.2 ± 0.6 °C; relative humidity [%rh] 47.5 ± 0.4%). During the trial, VO_2_ and RER were collected to calculate H_prod_, H_evap_, and H_dry_. Following the baseline trial, participants completed self-directed outdoor summer training followed by a post-HAz trial. Then, five days of HA were completed over eight days in the heat (T_amb_, 38.7 ± 1.1 °C; %rh, 51.2 ± 2.3%). During the HA sessions, participants exercised to maintain hyperthermia (38.50 °C and 39.75 °C) for 60 min. Then, a post-HA trial was performed. There were no differences in H_prod_ between the baseline (459 ± 59 W·m^−2^), post-HAz (460 ± 61 W·m^−2^), and post-HA (464 ± 55 W·m^−2^, *p* = 0.866). However, H_evap_ was significantly increased post-HA (385 ± 84 W·m^−2^) compared to post-HAz (342 ± 86 W·m^−2^, *p* = 0.043) and the baseline (332 ± 77 W·m^−2^, *p* = 0.037). Additionally, H_dry_ was significantly lower at post-HAz (125 ± 8 W·m^−2^, *p* = 0.013) and post-HA (121 ± 10 W·m^−2^, *p* < 0.001) compared to the baseline (128 ± 7 W·m^−2^). H_dry_ at post-HA was also lower than post-HAz (*p* = 0.049). H_prod_ did not change following HAz and HA. While H_dry_ was decreased following HA, the decrease in H_dry_ was smaller than the increases in H_evap_. Adaptations in body heat exchange can occur by HA following HAz.

## 1. Introduction

Greater physiological strain, such as an increased heart rate (HR) and internal body temperature, is placed on the body when an individual performs exercise in hot environments compared to exercising in temperate environmental conditions [1]. Additionally, exercise in the heat negatively impacts exercise performance and athlete safety [2]. For example, marathon performance progressively decreases as the Wet Bulb Globe Temperature (WBGT) increases from 5 °C to 25 °C [3]. Furthermore, exertional heat stroke is among the top three leading causes of death in sport, and other exertional heat illnesses, including heat exhaustion, heat syncope, and heat cramps, are prevalent and recurring illnesses across all levels of sport [4,5]. These negative implications occur when heat production is greater than heat dissipation.

Body heat storage is described by the human heat balance equation: S = M − Wk ± R ± C ± K − E (W), where S is the storage of heat within the human body, M is the metabolic heat production, Wk is the work rate (useful mechanical power) accomplished, C is convective heat loss to the environment, K is conductive heat loss to the environment, R is radiant heat loss to the environment, and E is evaporative heat loss from the skin to the environment [6]. Thermoregulation is achieved through a balance between heat production (M − Wk) and heat loss (R + C + K + E). Metabolic heat production (M − Wk) refers to the amount of heat, which is not used for work after being released as energy. Approximately 75–90% of the energy produced does not contribute to work performance, and it is liberated as heat, which leads to an increase in internal body temperature if heat production exceeds heat dissipation [7,8,9]. Sweat evaporation (E) plays an important role in thermoregulation, and the body is capable of dissipating 30–100% of metabolic heat through evaporation [10,11]. This avenue of heat loss is critical during exercise in the heat, especially in a hot dry environment [6]. In addition to evaporative cooling from sweat, dry heat loss (C, K, and R) is achieved through the transfer of heat from high (skin) to low temperature (environment). The balance between heat production and heat dissipation determines the internal body temperature.

Heat acclimation (HA) and heat acclimatization (HAz) are impactful strategies that mitigate physiological strain during exercise-heat stress [1]. HA refers to training in a hot artificial environment and HAz indicates training in an outdoor natural hot environment. Adaptations following HA and HAz include decreases in HR, internal body temperature, skin temperature (T_sk_), rating of perceived exertion, thermal sensation, sweat sodium, and chloride concentrations, in addition to increases in plasma volume, sweat rate, and skin blood flow [1,12]. While all adaptations are important, an adaptation of internal body temperature is critical to both reducing the risk of heat illness and increasing exercise performance. An increase in internal body temperature during exercise in the heat has been associated with higher HR and lower stroke volume, mean arterial pressure, and potentially cardiac output [13,14]. In addition to these negative physiological outcomes, higher internal body temperature has been reported to induce greater fatigue and reduce exercise performance in trained individuals [15].

Previous studies indicated that HA increases evaporative heat loss and maximal skin wetness, which leads to higher capacity of the body to dissipate heat and decreased dry heat loss [16,17]. However, these studies controlled metabolic heat production, which is another factor that impacts internal body temperature, due to the purposes of the studies. One previous study investigating metabolic heat production and evaporative heat loss following HA concluded that metabolic heat production decreased, but evaporative heat loss did not change in untrained individuals [18]. It has been demonstrated that the adaptations following HA between trained and untrained individuals are different [2,19]. In addition, it is important for trained individuals to understand changes in body heat exchange, including metabolic heat production. Additionally, when metabolic heat production is not controlled during exercise, the adaptations in evaporative heat loss and dry heat loss following HA might be different. Thus, the purpose of this study was to examine the changes in metabolic heat production, evaporative heat loss, and dry heat loss following HAz and HA to investigate the factors impacting adaptations of internal body temperature in trained endurance athletes.

## 2. Materials and Methods

### 2.1. Participants

Twenty-two male endurance athletes (mean [M] ± standard deviation [SD]; age, 37 ± 12 y; body mass, 73.4 ± 8.7 kg; height, 178.7 ± 6.8 cm; %body fat, 10.8 ± 5.2%; and VO_2max_, 57.1 ± 7.2 mL·kg^−1^·min^−1^) were recruited from the local running community and participated in this study. Following an explanation of study procedures, which were approved by the Institutional Review Board at the University of Connecticut where this study was conducted, participants provided written and informed consent to participate in this study. The current study was a part of larger research study; however, the research question examined in this study was different from the other research questions investigated. Other investigations were (1) the effects of heat acclimatization and heat acclimation on physiological variables, such as internal body temperature, HR, and sweat electrolyte, and (2) the effects of heat acclimatization and heat acclimation on a 4 km time trial, which were included in other research papers [20] (currently under review).

### 2.2. Procedure

First, participants completed a maximal oxygen consumption (VO_2max_) test with a graded running exercise on a standardized treadmill (T150; COSMED, Traunstein, Germany) to collect VO_2max_ and the velocity of VO_2max_ (vVO_2max_). The participants completed 5 min of a self-selected pace warmup before beginning the test. During the test, the speed was increased either 0.5 or 1.0 mile·h^−1^ after completing each 2 min stage, and participants continued exercising until reaching maximal volition.

Before participants received any heat exposure, they performed a baseline test to measure their physiological responses to the heat in the lab. The test consisted of 60 min of exercise at 59 ± 2% vVO_2max_ in the heat (M ± SD; ambient temperature [T_amb_], 35.2 ± 0.6 °C; relative humidity [%rh], 47.5 ± 0.4%; WBGT, 29.5 ± 0.6 °C; wind speed 4.0 ± 0.1 mile·h^−1^). Participants provided urine samples to measure their hydration status to ensure they started testing in a euhydrated status (M ± SD; urine specific gravity [USG], 1.010 ± 0.009; color, 2 ± 0) [21]. If the USG was above 1.020 and below 1.025, participants consumed 500 mL of water prior to the start of testing. During the test, rectal temperature (T_rec_) (YSI probe, MP160; BIOPAC Systems Inc., Goleta, CA, USA), HR (H10^®^, Polar Electro™, Kempele, Finland) and the mean T_sk_ (iButton; iButton Link LLC., Whitewater, WI, USA) of four sites including the thigh, chest, upper arm, and calf were measured [22]. Sweat rate was calculated based on pre and post exercise nude body mass. Participants did not consume any fluid during testing. Additionally, oxygen consumption (VO_2_) and respiratory exchange ratio (RER) were collected using a standard metabolic cart at 5–10, 30–35, and 55–60 min (TrueOne^®^ Metabolic Measurement System; PARVO MEDICS Inc., Sandy, UT, USA). A standard metabolic cart was calibrated before tests.

Following the baseline testing, participants performed self-directed summer training (early June–end of August). Training loads, such as total distance covered, training time, and average HR, and environmental conditions for each summer training session were monitored (Garmin, Garmin™ Ltd., Olathe, KS, USA; Polar H10, Polar Electro™, Kempele, Finland; Bryton Rider, Bryton™ Inc., Taipei City, Taiwan). Exercise types during self-selected summer training included anything that participants normally performed, such as running and cycling. After summer training (HAz, 108 ± 10 days), participants performed the same test (post-HAz) followed by 5 days of HA sessions in the heat (M ± SD; T_amb_, 38.7 ± 1.1 °C; %rh, 51.2 ± 2.3%; and WBGT, 33.8 ± 1.1 °C). During the HA sessions, participants completed running exercises to achieve hyperthermia for 60 min. Hyperthermia was defined as T_rec_ between 38.50 °C and 39.75 °C, whose HA induction method is defined as “hyperthermic zone HA” (HZHA). Participants begun HA sessions with a higher exercise intensity (~70% vVO_2max_) to increase T_rec_ rapidly to 38.5 °C and continued to exercise remaining 60 min with adjusted intensity to maintain T_rec_ in the hyperthermic zone.

### 2.3. Heat Exchange Calculations

Partitional calorimetry was used to calculate heat exchanges. Metabolic heat production (H_prod_) was calculated by subtracting the rate of external work performed (running) (Wk) from the concurrent rate of metabolic energy expenditure divided by surface area (A_D_) for each participant (M) [6]:H_prod_ (W·m^−2^) = (M − Wk)/(A_D_)
$$M=VO_{2}\cdot \frac{\left\{\left[\left(\frac{RER-0.7}{0.3}\cdot 21.13\right)+\left[\left(\frac{1.0-RER}{0.3}\right)\cdot 19.62\right]\right]\right\}}{60}\cdot 1000(W)$$

The average rate of evaporative heat loss from the skin surface (H_evap_) and sweat efficiency (S_eff_) were calculated using the following equations [6]: $$H_{evap}\left(W\right)=Whole\;body\;sweat\;rate\cdot 2426\cdot \frac{s_{eff}}{60}$$
$$s_{eff}=1-\frac{\omega^{2}req}{2}$$

Sweat efficiency was calculated by the skin wettedness (ω_req_) required for heat balance [6]. ω_req_ was calculated by using the total evaporation required to maintain heat balance at zero (E_req_) and the rate of maximal evaporation when the skin was completely wet (E_max_) [6]. If ω_req_ > 1, a sweat efficiency value of 1 was assumed [23]. Dry heat exchange at the skin surface (H_dry_) consists of convection (C_skin_), radiation (R_skin_), and conduction (K_skin_) [6]. Mean radiant temperature was 0 °C in the current study since testing was performed in an environmentally controlled room. For the area weighted emissivity of the clothed body surface, the effective radiative area of the body, and insulation value were 0.98, 0.73, and 0.10, respectively. Respiratory heat loss (H_res_) is achieved through convection and evaporation [6]. The vapor pressure at the skin surface when saturated with sweat (P_skin,sat_), the partial pressure of water vapor in ambient air (P_a_), the evaporative resistance of clothing (R_e,cl_), the evaporative heat transfer coefficient (h_e_), and the clothing area factor (f_cl_) are all involved in E_max_ [6]. Additionally, running economy (RE), which is referred to as the amount of oxygen utilized at the given exercise intensity, was measured by VO_2_. More detailed information about partitional calorimetry is described in the previous study [6].
$$\omega_{req}=\frac{E_{req}}{E_{\max}}$$

E_req_ (W) = H_prod_ − H_dry_ − H_res_

H_dry_ = C_skin_ + R_skin_ + K_skin_


$$E_{\max}=\frac{\left(P_{skin,sat}-P_{a}\right)}{\left(R_{e,cl}+\frac{1}{h_{e}\cdot f_{Cl}}\right)}\cdot A_{D}$$


Repeated measures ANOVAs with LSD pairwise comparisons were performed to assess differences between mean H_prod_, H_evap_, H_dry_, E_req_, E_max_, and RE at baseline, post-HAz, and post-HA. Effect sizes (ES) were calculated using Hedges’ g with the resulting effects identified as either small (0.2–0.49), medium (0.5–0.79), or large (>0.8) effects [24]. Data are reported as M ± SD, 95% confidence intervals (95%CI) and ES. Stepwise linear regression was used to predict maximum T_rec_ during the 60 min tests from H_prod_, H_evap_, H_dry_, T_sk_, sweat rate, and minimum T_rec_, which was the lowest T_rec_ during the tests. All statistical analyses were completed using SPSS Statistics for Mac, version 25 (IBM Corp., Armonk, NY, USA). Significance was set at *p* ≤ 0.05.

## 3. Results

### 3.1. HAz and HA Induction

During HAz, the average training duration at each session was 58.36 ± 78.50 min for running and 94.36 ± 70.94 min for cycling. The average HR was 138.60 ± 14.76 bpm for running and 128.06 ± 15.93 bpm for cycling during HAz. The average WBGT of all HAz sessions was 22.26 ± 4.31 °C for running and 23.72 ± 4.03 °C for cycling. The duration, average rectal temperature (T_rec_), average T_rec_ for the hyperthermia period, average heart rate (HR), and the average HR for the hyperthermia period during heat acclimation from Day 1 to Day 5 are described in Table 1.

### 3.2. Metabolic Heat Production, Evaporative Heat Loss, and Dry Heat Loss

The results regarding H_prod_, H_evap_, and H_dry_ are described in Table 2. There were no differences in H_prod_ between the baseline, post-HAz, and post-HA (*p* = 0.866) (Figure 1). However, H_evap_ was significantly increased at post-HA compared to post-HAz and the baseline (Figure 2). Additionally, H_dry_ was significantly lower at post-HAz and post-HA compared to the baseline (Figure 3). Furthermore, H_dry_ at post-HA was lower than post-HAz. A larger percentage of heat dissipation occurred through H_evap_ at post-HA (M ± SD; 75.4 ± 4.8%) compared to the baseline (M ± SD; 71.4 ± 5.0%, *p* = 0.004) and post-HAz (M ± SD; 72.2 ± 6.0%, *p* = 0.013), while there were no differences between the baseline and post-HAz (*p* = 0.468). Moreover, a smaller percentage of heat dissipation occurred through H_dry_ at post-HA (M ± SD; 24.6 ± 4.8%) compared to the baseline (M ± SD; 28.6 ± 5.0%, *p* = 0.004) and post-HAz (M ± SD; 27.8 ± 6.9%, *p* = 0.013), while there were no differences between the baseline and post-HAz (*p* = 0.468).

### 3.3. Running Economy, and E_max_, and E_req_

Data for RE, E_max_, and E_req_ are demonstrated in Table 2. RE and E_req_ were unchanged at baseline, post-HAz, and post-HA (E_req_, *p* = 0.921; RE, *p* = 0.441). However, E_max_ increased at post-HA compared to post-HAz. 

### 3.4. Body Heat Exchange and Physiological Responses

Only metabolic H_prod_ predicted a maximum T_rec_ (r^2^ = 0.192, *p* = 0.041) among H_prod_, H_evap_, H_dry_, T_sk_, sweat rate, and minimum T_rec_ at the baseline. However, lower T_sk_, minimum T_rec_, and H_prod_ significantly predicted a lower maximum T_rec_ with r^2^ = 0.502 (*p* < 0.001) from the results of post-HAz and post-HA. Furthermore, decreases in T_sk_ alone significantly predicted a lower maximum T_rec_ (r^2^ = 0.301 *p* < 0.001).

## 4. Discussion

The purpose of this study was to examine the changes in metabolic heat production, evaporative heat loss, and dry heat loss following HAz and HA in endurance athletes. The current study found that there was no change in metabolic heat production following HAz and HA. However, evaporative heat loss increased and dry heat loss decreased, and the greater percentage of heat dissipation relied on evaporation following HA. These findings add to the current literature that helps to understand the principal mechanism of adaptations in body heat exchanges following HA and HAz in endurance athletes, especially how internal body temperature decreases after multiple heat exposures. Investigating whether HA elicited adaptations in body heat exchange following HAz was a unique approach since most athletes train outside, and HA following summer training could induce additional adaptations with a shorter HA.

Metabolic heat production has been identified as one of the determining factors influencing internal body temperature during exercise [6]. In the current study, there was no difference in metabolic heat production following HA in trained athletes. This finding conflicts with previous findings, which demonstrated metabolic heat production decreased following HA in untrained athletes [18]. This previous study concluded that lower metabolic heat production was determined to be the primary factor for induction of lower internal body temperature following HA, even with documented improvements in running economy [18]. There was the possibility that metabolic heat production might be reduced at the given exercise intensity due to an improvement in running economy. In the current study, participants were trained endurance athletes and, therefore, did not demonstrate improvements in running economy following HA, similar to the conclusions of previous investigations [25]. Therefore, changes in metabolic heat production were not observed and were not the determining factor that induced lower T_rec_ in trained endurance athletes following HA.

In contrast, evaporative heat loss was increased following HA in the current study, which could indicate it may be the main factor for achieving lower T_rec_ following HA. Interestingly, similar responses for evaporative heat loss following HA were not previously reported for untrained participants [18]. In the previous study, participants were observed to reach a plateau for internal body temperature during exercise, which indicated the amount of heat dissipation was matched to heat production. Thus, evaporative heat loss did not increase following HA, while internal body temperature was decreased due to lower metabolic heat production [18]. In the current study, evaporative heat loss was improved, which induced lower T_rec_. This result was supported by a previous study indicating an increase in evaporative heat loss following HA when metabolic heat production was controlled during exercise [16]. The current study did not find the change in evaporative heat loss following HAz, and this could be due to the environmental conditions during self-directed summer training. Thus, increased evaporative heat loss was the primary factor inducing a lower T_rec_ following HA in endurance trained athletes, which is known to decrease fatigue and increase exercise performance in the heat [15].

A lower T_sk_, minimum T_rec_, and metabolic heat production significantly predicted a lower maximum T_rec_ (r^2^ = 0.502) from the results of the post-HAz and post-HA. Additionally, lower T_sk_ alone significantly predicted a lower maximum T_rec_ (r^2^ = 0.301). The evaporation of sweat from the skin surface results in a cooling effect of 2426 J·g^−1^ [26]. A lower T_sk_ subsequently decreases redistribution of blood to the cutaneous circulation, which receives up to 50–70% of cardiac output during heat stress [26,27]. Increasing cardiac output helps to deliver oxygen to exercising muscles, which helps to increase exercise performance [13]. 

The dry heat loss was also observed to decrease following HA in the current study. One potential mechanism of explanation could be the resulting lower T_sk_ induced by greater evaporative heat loss. Lowering T_sk_ has been observed to decrease the gradient between the skin surface and the ambient air, which led to a smaller amounts of dry heat loss following HA [16]. However, the rate of decrease in dry heat loss following HA was smaller compared to the rate of increase in evaporative heat loss, thus, T_rec_ was decreased following HA.

This study was not without limitations. While the indirect measurements of metabolic heat production, evaporative heat loss, and dry heat loss are widely used and well accepted in previously conducted studies, the most accurate method to determine whole-body evaporative and dry heat exchange is direct calorimetry [6,16,27]. However, this method is typically not feasible as it requires the complete evaporation of all sweat from the skin and is typically performed in a calorimeter with a high and turbulent air flow [28]. Another limitation of this study was that heat loss and dry heat loss are dependent on the environmental testing conditions, as well as the method of HA induction. For example, evaporative heat loss is limited when exercise is performed in high humidity; therefore, dry heat loss is the primary mechanism to dissipate heat in this case. Additionally, it has been previously demonstrated that a greater adaptation in sweat rate was achieved following HA in dry conditions compared to humid conditions [29]. Moreover, while the findings from the current study apply to trained endurance athletes, studies encompassing other populations may provide different results. Furthermore, while everyone started testing in a euhydrated state, fluid loss was not replaced during testing, and the dehydration level at the end of testing was greater following HA due to the higher sweat rate. Dehydration can impair the sweat rate; therefore, if fluids were replaced to compensate for the amount of sweat lost, then evaporative heat loss could be even higher following HA [30].

## 5. Conclusions

In conclusion, metabolic heat production was not changed; however, evaporative heat loss was increased following HA. While dry heat loss was decreased due to the lower T_sk_ following HA, the rate of change was smaller compared to the increase in evaporative heat loss. Moreover, a larger percentage of heat dissipation relied on evaporative heat loss following HA. Thus, the primary factor to induce lower T_rec_ following HA was the adaptation of an increased evaporative heat loss in trained endurance athletes. Thus, it is critical to achieve a greater amount of hyperthermia and sweat during HA induction to induce this adaptation. The factors influencing adaptations in sweat, such as exercise intensity, duration, and environmental conditions, are considered important when creating an optimal HA induction protocol.

## Figures and Tables

**Figure 1 ijerph-19-06412-f001:**
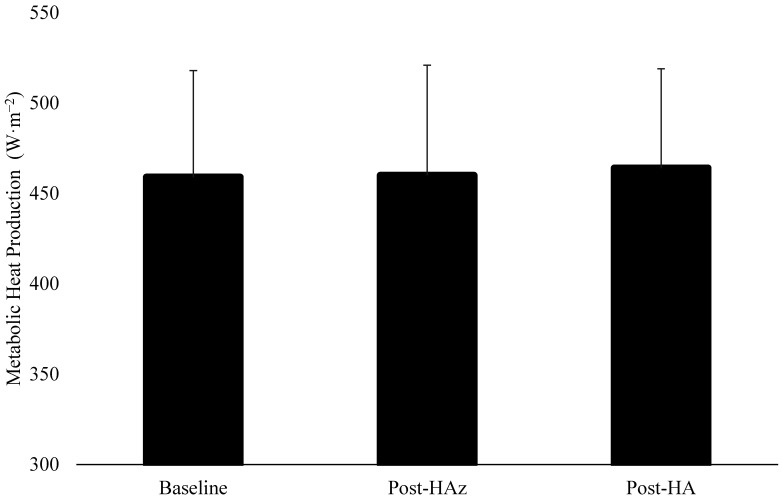
Changes in metabolic heat production following heat acclimatization and acclimation. Baseline indicates unacclimatized, Post-HAz indicates post-heat acclimatization, Post-HA indicates post-heat acclimation.

**Figure 2 ijerph-19-06412-f002:**
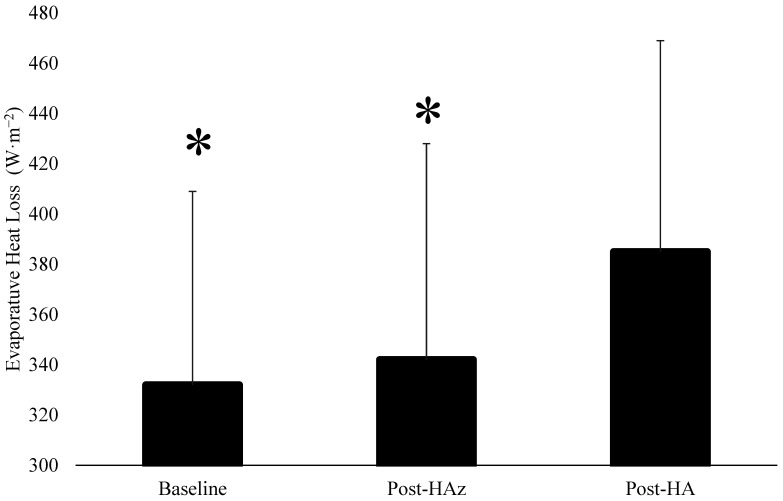
Changes in evaporative heat loss following heat acclimatization and acclimation. * indicates statistical significance following heat acclimation (Post-HA), *p* ≤ 0.05. Baseline indicates unacclimatized, Post-HAz indicates post-heat acclimatization, Post-HA indicated post-heat acclimation.

**Figure 3 ijerph-19-06412-f003:**
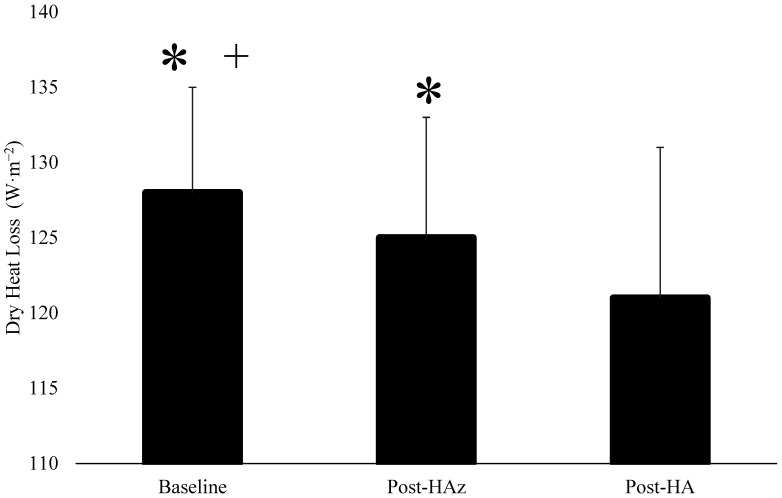
Changes in dry heat loss following heat acclimatization and acclimation. * indicates statistical significance following heat acclimation (Post-HA), and + indicates statistical significance following heat acclimatization (Post-HAz), *p* ≤ 0.05. Baseline indicates unacclimatized, Post-HAz indicates post-heat acclimatization, Post-HA indicates post-heat acclimation.

**Table 1 ijerph-19-06412-t001:** The duration, average rectal temperature (T_rec_), average T_rec_ for hyperthermia period, average heart rate (HR), and average HR for hyperthermia period during heat acclimation from Day 1 to Day 5.

	Day 1	Day 2	Day 3	Day 4	Day 5
Duration (min)	82 ± 6	81 ± 6	84 ± 6	83 ± 8	84 ± 5
Ave T_rec_ (°C)	38.89 ± 0.40	38.94 ± 0.31	38.84 ± 0.40	38.82 ± 0.31	38.78 ± 0.32
Ave T_rec_ for hyperthermia (°C)	39.22 ± 0.37	39.24 ± 0.21	39.19 ± 0.38	39.20 ± 0.29	39.13 ± 0.23
Ave HR (bpm)	136.33 ± 12.94	132.34 ± 13.66	131.63 ± 12.03	131.00 ± 12.68	129.27 ± 12.42
Ave HR for hyperthermia (bpm)	137.35 ± 13.99	132.33 ± 13.70	130.66 ± 14.12	132.31 ± 13.89	128.25 ± 12.56

The maximum HR, maximum T_rec_, and T_sk_ were significantly lower at post-HA (M ± SD; HR, 149.55 ± 15.63 bpm; T_rec_, 38.77 ± 0.52 °C; and T_sk_ 35.52 ± 0.60) compared to both the baseline (M ± SD; HR, 161.91 ± 15.28 bpm, *p* < 0.001; T_rec_, 39.16 ± 0.53 °C, *p* = 0.002; and T_sk_, 36.27 ± 0.49, *p* < 0.0001) and post-HAz (M ± SD; HR, 155.73 ± 17.68 bpm, *p* = 0.003; T_rec_, 39.03 ± 0.51 °C, *p* = 0.012; and T_sk_, 35.93 ± 0.50, *p* = 0.001). Moreover, HR (*p* = 0.013) and T_sk_ (*p* = 0.001) were significantly lower at post-HAz compared to the baseline.

**Table 2 ijerph-19-06412-t002:** Metabolic heat production, evaporative heat loss, dry heat loss, running economy, rate of maximal evaporative heat loss, and total evaporation required for heat balance of male endurance athletes. Baseline indicates unacclimatized, Post-HAz indicates post-heat acclimatization, Post-HA indicates post-heat acclimation (dual heat acc). Data are presented as mean (M) ± standard deviation (SD), effect size (ES), 95% confidence intervals (95%CI). * indicates statistical significance, *p* ≤ 0.05.

Metabolic Heat Production (W·m^−2^)							
Test	M ± SD		Test	M ± SD	ES	95%CI	*p*-Value
Baseline	459 ± 59	vs.	Post-HAz	460 ± 61	0.02	−21, 21	0.975
			Post-HA	464 ± 55	0.09	−27, 17	0.649
Post-HAz	460 ± 61	vs.	Post-HA	464 ± 55	0.07	−25, 16	0.649
**Evaporative Heat Loss (W·m^−2^)**							
**Test**	**M** **±** **SD**		**Test**	**M** **±** **SD**	**ES**	**95%CI**	** *p* ** **-Value**
Baseline	332 ± 77	vs.	Post-HAz	342 ± 86	0.12	−40, 20	0.486
			Post-HA	385 ± 84	0.66	−103, −4	0.037 *
Post-HAz	342 ± 86	vs.	Post-HA	385 ± 84	0.51	−85, −1	0.043 *
**Dry Heat Loss (W·m^−2^)**							
**Test**	**M** **±** **SD**		**Test**	**M** **±** **SD**	**ES**	**95%CI**	** *p* ** **-Value**
Baseline	128 ± 7	vs.	Post-HAz	125 ± 8	0.40	0.8, 5.8	0.013 *
			Post-HA	121 ± 10	0.81	3.8, 11.2	<0.001 *
Post-HAz	125 ± 8	vs.	Post-HA	121 ± 10	0.44	0.01, 8.4	0.049 *
**Running Economy (ml** **·** **kg** ** ^−^ ** ** ^1^ ** **·** **min** ** ^−^ ** ** ^1^ ** **)**							
**Test**	**M** **±** **SD**		**Test**	**M** **±** **SD**	**ES**	**95%CI**	** *p* ** **-Value**
Baseline	37.0 ± 5.1	vs.	Post-HAz	37.0 ± 4.8	0.00	−1.7, 1.8	0.980
			Post-HA	37.3 ± 4.1	0.06	−2.1,1.5	0.746
Post-HAz	37.0 ± 4.8	vs.	Post-HA	37.3 ± 4.1	0.07	−2.0, 1.4	0.710
**Rate of maximal evaporation, E_max_ (W·m^−2^)**							
**Test**	**M** **±** **SD**		**Test**	**M** **±** **SD**	**ES**	**95%CI**	** *p* ** **-Value**
Baseline	281 ± 16	vs.	Post-HAz	272 ± 26	0.42	−2.7, 21.1	0.123
			Post-HA	288 ± 19	0.40	−15.6, 1.6	0.106
Post-HAz	272 ± 26	vs.	Post-HA	288 ± 19	0.70	−27.4, -5.0	0.007 *
**Total evaporation required for heat balance, E_req_ (W·m^−2^)**							
**Test**	**M** **±** **SD**		**Test**	**M** **±** **SD**	**ES**	**95%CI**	** *p* ** **-Value**
Baseline	306 ± 57	vs.	Post-HAz	310 ± 58	0.07	−23.4, 15.8	0.690
			Post-HA	318 ± 50	0.22	−33.2, 8.7	0.237
Post-HAz	310 ± 58	vs.	Post-HA	318 ± 50	0.15	−28.5, 11.6	0.391

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
