# Peer review of "Effects of Heat Acclimation Following Heat Acclimatization on Whole Body Heat Exchange in Trained Endurance Athletes"

_ijerph, 2022, doi:10.3390/ijerph19116412_

Round 1

Reviewer 1 Report

ABSTRACT: Significance of findings is missing

INTRODUCTION:

Please consider including a more recent reference for the EHI position statement in lines 39-42. an example is:

Casa DJ, DeMartini JK, Bergeron MF, Csillan D, Eichner ER, Lopez RM, Ferrara MS, Miller KC, O'Connor F, Sawka MN, Yeargin SW. National Athletic Trainers' Association position statement: exertional heat illnesses. Journal of athletic training. 2015 Sep;50(9):986-1000.

METHODS:

Please state how the 22 male endurance trained athletes were recruited.

Line 97 - change "examining" to "examined"

Line 98 - change "investigating" to "investigated in"

There was no specific reference to cycling in the methods

DISCUSSION:

Please include unique strengths of your study approach (if any)

REFERENCES:

DOI is missing in references 4, 9, 11, 19, & 22

Reviewer 2 Report

In my opinion, the work has relevant results, which are well discussed and presented. However, the text needs some improvements, mainly in detailing the methods.

major issues

Heat exhange calculations section - The description of the heat balance lacks details that could make the methodology reproducible. For example, how convection, radiation and energy transfer by respiration were calculated. What are the values ​​of the convection and evaporation coefficients and emissivity. Which parameters were used for clothing conduction and its area factor. How the work of running or cycling were estimated.

l.168 How is the standard error used in tests defined? Were instrument uncertainties considered? Were the instruments calibrated before testing? What are your uncertainties?

minor issues

l.44 - In the body's energy balance, there isn't the term of respiration heat loss, as presented in the methods.

l.98 - is repeating that it is part of a large study.

l. 117 - Which sensor was used to measure rectal temperature. If I'm not mistaken, only the acquisition system was mentioned.

Make clear the types of exercises performed in the methods. The results discuss running and cycling, but it is unclear in the methods section.

l. 175 I found it peculiar to have a greater variation than the average.

l. 177 Average WBGT at each HAz session - isn't it the average of all sections?

l. 179-193 I think it would be easier to follow this data if it were in a table.

Round 2

Reviewer 2 Report

In my opinion, all the issues raised were answered and the article had enough improvements for its publication.